# Comparative Assessment of the Anti-*Helicobacter pylori* Activity and Gastroprotective Effects of Three Herbal Formulas for Functional Dyspepsia In Vitro

**DOI:** 10.3390/cells13110901

**Published:** 2024-05-24

**Authors:** Jing-Hua Wang, Song-Yi Han, Jisuk Kim, Sookyoung Lim, Chaehee Jeong, Liangliang Wu, Hojun Kim

**Affiliations:** Department of Rehabilitation Medicine of Korean Medicine, Dongguk University, 814 Siksa-dong, Ilsandong-gu, Goyang-si 10326, Republic of Korea; ewccwang@gmail.com (J.-H.W.); syh12156@gmail.com (S.-Y.H.); kimjikimji730@gmail.com (J.K.); sklim1972@naver.com (S.L.); chaeh305@naver.com (C.J.); dbnbn8@gmail.com (L.W.)

**Keywords:** herbal formula, *Helicobacter pylori*, antimicrobial effect, functional dyspepsia, gastritis

## Abstract

*Helicobacter pylori* has been implicated in various gastrointestinal disorders, including functional dyspepsia. This study aimed to compare the anti-*H. pylori* activity and gastroprotective effects of three typical herbal formulas used for gastrointestinal disorders in Korea: Shihosogan-tang (ST), Yijung-tang (YT), and Pyeongwi-san (PS). Firstly, we assessed the total phenolic and flavonoid contents, as well as the antioxidative capacity. Additionally, we evaluated the antibacterial effect on *H. pylori* using an ammonia assay, minimum inhibitory concentration (MIC) test, and the disk agar diffusion method. Furthermore, we examined alterations in the gene expression of tight junction proteins, pro-inflammatory cytokines, and cellular vacuolation using an AGS cell model infected with *H. pylori*. While ST exhibited a higher total phenolic content, superior free radical scavenging, and inhibition of *H. pylori* compared to YT and PS, YT more evidently inhibited gastric cellular morphological changes such as vacuolation. All formulations significantly ameliorated changes in inflammatory and gastric inflammation-related genes and cellular morphological alterations induced by *H. pylori* infection. Overall, the present in vitro study suggests that all three herbal formulas possess potential for ameliorating gastrointestinal disorders, with ST relatively excelling in inhibiting *H. pylori* infection and inflammation, while YT potentially shows greater efficacy in directly protecting the gastric mucosa.

## 1. Introduction

Functional dyspepsia (FD) presents as a challenging gastrointestinal condition primarily impacting upper digestive tract motility, characterized by the absence of structural abnormalities [1]. Large population-based studies indicate a global prevalence ranging from approximately 11.5% to 29.2% [2], notably higher in Asian countries compared to Western nations [3]. Treatment options for FD encompass various drug classes, such as proton pump inhibitors (PPIs), H2 receptor antagonists, prokinetic agents, antidepressants, and antacids [4]. However, the therapeutic efficacy for FD is often insignificant or challenging to sustain due to its unclear etiology [5]. Consequently, the U.S. Food and Drug Administration has not approved any specific FD treatment medications [6].

*Helicobacter pylori* is a critical pathogenic bacterial species that colonizes the stomach, causing various gastrointestinal disorders such as peptic ulcers, gastritis, and gastric cancer [7]. It is estimated that over 3 billion people worldwide are infected with *H. pylori*, with the majority of cases found in developing countries [8]. Epidemiological studies report the prevalence of *H. pylori* infection in individuals with functional dyspepsia to range from approximately 30% to 70% [9]. Despite some controversy, previous clinical findings have shown that eradicating *H. pylori* leads to symptom improvement in about 70% of infected patients [10]. Furthermore, ample evidence suggests that *H. pylori* triggers gastric acid secretion, affects motility, influences the immune response, and alters the gut microbiome, contributing to functional dyspepsia [11]. However, eradicating *H. pylori* with antibiotics faces numerous challenges, including drug resistance, side effects, and individual variations [12]. Consequently, identifying an effective and safe therapy to inhibit *H. pylori* infection could be a promising strategy for alleviating gastrointestinal disorders, including *H. pylori*-associated functional dyspepsia.

Oxidative stress and inflammation are common pathological factors associated with gastrointestinal motility [13]. *H. pylori* infection of gastric epithelial cells results in the generation of numerous free radicals, triggering inflammatory reactions [14]. This inflammation can disrupt gut motility and lead to functional disorders [15]. Therefore, many scientists and physicians consider alleviating oxidative stress and inflammation as potential therapeutic targets for improving various gastrointestinal disorders [13].

Recently, numerous herbal medicines, long utilized in traditional practices, have shown promise in treating various gastrointestinal disorders, as evidenced by both clinical and preclinical studies [16,17,18]. In this investigation, we focused on three commonly used oriental herbal formulas covered by the National Health Insurance Service (NHIS, www.nhis.or.kr, accessed on 14 February 2024): Shihosagan-tang (ST, originating from “Jing-Yue-Quan-Shu” in 1624 A.D.), Ijung-tang (YT, originating from “Shang-Han-Lun” in 219 A.D.), and Pyeong-wi-san (PS, originating from Dong-Eui-Bo-Gam in 1613 A.D.). According to traditional medical theory, ST is primarily used for treating liver Qi stagnation, YT is mainly employed for asthenic symptoms, and PS is largely utilized to address food impaction. Collectively, these formulas are renowned for their clinical efficacy in addressing diverse gastrointestinal ailments. Nevertheless, it remains unclear which formula exhibits a more effective gastroprotective effect and stronger antimicrobial activity against pathogens such as *H. pylori*.

Hence, we undertook a comparative assessment of the anti-*H. pylori* activity and gastroprotective effects of these three renowned herbal formulas within an *H. pylori*-infected gastric epithelial cell model to establish a foundation for in vivo studies and clinical applications. Gaining insight into the relative efficacy of these herbal formulations could serve as a guiding principle for future research endeavors and inform clinical practices concerning their utilization.

## 2. Materials and Methods

### 2.1. Extraction and Preparation of Herbal Formulas

The herbal formulas ST, YT, and PS, as per the Korean Pharmacopoeia standards, were procured from Dongguk University Ilsan Medical Center (Goyang, Gyeong-gi-do, Republic of Korea). To ensure a standardized comparison and sufficient acquisition of both fat-soluble and water-soluble compounds, we opted to use the 30% ethanol extract method for the three herbal formulas. Detailed herbal prescriptions and standard criteria are provided in Figure 1A. Each 50 g of powder was ground and then individually extracted with 500 mL of 30% ethanol (Merck, Rahway, NJ, USA) by heating at 100 °C for 60 min. After centrifugation at 3000× *g* for 10 min, the supernatants were filtered through Grade 4 filter paper (Whatman, Kent, UK). The collected extract was evaporated using a rotary evaporator (EYELA N-1200A, Tokyo, Japan) and lyophilized using a vacuum freeze dryer (Bondiro, IlshinBioBase, Gyeonggi-do, Republic of Korea) at −70 °C, resulting in yields of 24.8% for ST, 35.1% for YT, and 36.1% for PS, respectively (*w/w*, Figure 1B). The obtained powders were stored at −20 °C. Prior to the in vitro determinations, the samples were reconstituted in phosphate-buffered saline (PBS) and then filtered through 0.45 µm syringe filters (Whatman™, Marlborough, MA, USA). All the concentrations of the herbal formulas mentioned in the subsequent sections refer to the final concentrations.

### 2.2. Determination of Total Phenolic and Flavonoid Content

The total phenolic content of the three herbal formulations was evaluated using the Folin–Ciocalteu colorimetric method [19]. Initially, 100 µL of each herbal sample was mixed with 2 mL of 2% Na_2_CO_3_ solution and incubated at room temperature (RT) for 3 min. Subsequently, 100 µL of 2N Folin–Ciocalteu reagent was introduced, and the mixture was left to incubate at 25 °C for 30 min. Absorbance was then recorded at 750 nm using a microplate reader (TECAN Spark reader; Greenmate Biotech Co., Männedorf, Switzerland). Total phenolic compounds were quantified as gallic acid equivalents (GAEs) per unit weight of the drug.

For the determination of total flavonoid content in the three drug samples, a previously established method was followed [20]. Initially, 10 μL of each sample was dissolved in 30 μL of 5% NaNO_2_ and allowed to stand for 5 min. Subsequently, 30 μL of 10% AlCl_3_ was added to the mixture and incubated at RT for 6 min. Finally, 100 μL of 1 N NaOH was introduced, and absorbance was measured at 415 nm using a spectrophotometer (TECAN Spark reader; Greenmate Biotech Co., Männedorf, Switzerland). The total flavonoid content was determined based on the catechin standard curve.

### 2.3. Determination of Antioxidative Capacity

The antioxidative capacities of the three drugs were compared using the DPPH (2,2-diphenyl-1-picrylhydrazyl) assay. A 0.2 mM solution of DPPH was prepared in ethanol. Subsequently, 20 μL of each herbal formula (100 μg/mL) was combined with 80 μL of the DPPH solution in a 96-well plate. The mixture was then incubated at RT for 30 min in darkness, and absorbance was measured at 517 nm. Ascorbic acid (100 μg/mL) served as the positive control.

The ABTS^+^ radical solution was generated by mixing 7.4 mM ABTS and 2.6 mM potassium persulfate in equal volumes. Then, 20 μL of each sample (100 μg/mL) was mixed with 200 μL of the prepared ABTS^+^ solution in a 96-well plate. The mixture was incubated at RT for 60 min in darkness, and the scavenging activities of the samples were determined by measuring absorbance at 760 nm.

### 2.4. H. pylori Strain and AGS Cell Culture

The *H. pylori* strain KCCM40449 (ATCC 43526, CagA^+^, VacA^+^) was obtained from the Korean Culture Center of Microorganisms (Seoul, Republic of Korea). *H. pylori* was cultured on 5% sheep blood agar (Trypticase Soy Agar with 5% Sheep Blood, BD, Sparks, MD, USA) under microaerophilic conditions at 37 °C in a 10% CO_2_ incubator (Thermo 3111; Thermo Fisher Scientific, MA, USA). Subsequently, *H. pylori* was cultured in Brucella broth (BD BBL, Sparks, MD, USA) containing 10% fetal bovine serum (FBS) under microaerophilic conditions for 72 h.

AGS cells from a human gastric adenocarcinoma epithelial cell line (ATCC, CRL-1739, Manassas, VA, USA) were cultured in RPMI-1640 (LM011–03, Welgene, Gyeongsan, Republic of Korea) supplemented with 10% FBS and 1% penicillin–streptomycin mixture. The cells were maintained at 37 °C in a 5% CO_2_ environment.

### 2.5. Assessment of Urease Activity in H. pylori Using Ammonia Assay

*H. pylori* was cultivated in Brucella broth with varying concentrations of ST, YT, and PS (12.5, 25, 50 mg/mL) in a 96-well plate under microaerophilic conditions at 37 °C in a 10% CO_2_ incubator (Thermo 3111, Waltham, MA, USA). Following a 72-h incubation period, 100 μL of culture medium was collected from each well and transferred to a clean 1.5 mL tube. The samples were centrifuged at 2500× *g* for 10 min to obtain the supernatant. To each supernatant, 5 μL of 20% urea was added and incubated at 37 °C for 10 min. In accordance with a previous method [21], urease activity was indirectly assessed by measuring ammonia levels using an Asan Set Ammonia Kit (Asan Pharmaceutical, Seoul, Republic of Korea). Briefly, 400 μL of deproteinization solution was added to each sample and centrifuged at 2500× *g* for 5 min. Next, 100 μL of supernatant was mixed with 100 μL of phenol (40 mg/mL), 50 μL of NaOH (35.6 mg/mL), and 100 μL of sodium hypochlorite (10%, *w*/*v*), followed by incubation at 37 °C for 10 min. Absorbance was measured at 630 nm.

### 2.6. Antibacterial Activity Assessment

The paper disc agar diffusion assay was conducted according to Lai et al. [22]. Initially, *H. pylori* was prepared at a concentration of 0.1 absorbance at OD600 (optical density at 600 nm), and 500 μL of the strain culture was spread on Brucella agar, followed by placing a paper disc (ø 8 mm, Whatman™, Clifton, NJ, USA). ST, YT, and PS samples were adjusted to concentrations of 50, 100, 200, and 400 mg/mL. Subsequently, 40 μL of each sample was applied to individual discs and incubated for 72 h at 37 °C in a 10% CO_2_ environment, after which inhibition zones were measured. AMX (amoxicillin) at concentrations of 0.05, 0.1, 0.5, and 1 μg/mL was used as a positive control for comparison.

To determine the minimum inhibitory concentration (MIC), *H. pylori* (5 × 10^6^ CFU/mL) was mixed with ST, YT, and PS (12.5, 25, 50, 100 mg/mL) or amoxicillin (0.25, 0.5, 1, 2, 4 μg/mL). Next, 10 µL of each mixture was immediately applied to Brucella agar plates. After 72 h of incubation, the MIC levels of ST, YT, and PS were comparatively evaluated in the absence of visible *H. pylori* colonies on the agar plates.

### 2.7. Cell Viability Assessment

The cell viability of the AGS cells was assessed using the D-plus CCK Cell Viability Assay Kit (CCK-3000, Dongin LS, Seoul, Republic of Korea) [23]. AGS cells were seeded at a density of 1 × 10^5^ cells/well in 96-well culture plates. After a 24 h incubation period, samples at concentrations of 25, 50, and 100 μg/mL were added to each well, followed by further incubation at 37 °C for 6 h. *H. pylori* cells were treated to achieve a multiplicity of infection (MOI) of 50 (5 × 10^6^ CFU/well), then incubated for an additional 24 h. Subsequently, 10 µL of D-plus CCK reagent was introduced to each well. After a 1 h incubation at 37 °C, the absorbance of the cell supernatant was measured at 450 nm. Cell viability was calculated using the following equation: (absorbance of sample/absorbance of control) × 100%.

### 2.8. Establishment of H. pylori-Infected AGS Cell Model and Morphological Evaluation

AGS cells were seeded at a density of 1 × 10^6^ cells/well in 6-well plates and cultured for 24 h in complete growth medium at 37 °C and 5% CO_2_. Subsequently, 10 µL of each extract was added and incubated for 6 h, resulting in a final drug concentration of 100 μg/mL. Finally, AGS cells were infected with *H. pylori* at an MOI of 100 (1 × 10^8^ CFU/mL) for 24 h. Cell morphology was observed and photographed using an optical microscope at 200× magnification (Olympus BX61, Tokyo, Japan). Vacuolated cells were counted in five random fields for each sample.

### 2.9. Real-Time PCR Analysis of Inflammation-Associated and Tight Junction Genes

AGS cells were seeded in a 6-well plate at a density of 1 × 10^6^ cells/well and cultured at 37 °C under 5% CO_2_. After 24 h, the cells were pretreated with ST, PS, or YT (100 μg/mL) for 6 h, followed by stimulation with *H. pylori* at an MOI of 100 (1 × 10^8^ CFU/mL) for 24 h. Total RNA was extracted using TRIzol reagent (Invitrogen, Carlsbad, CA, USA) and reverse-transcribed with AccuPower RT Premix (BIONEER Corporation, Daejeon, Republic of Korea). Quantitative real-time polymerase chain reaction (qPCR) was then conducted using SYBR Green master mix (QPK-201; Toyobo, Tokyo, Japan) on a LightCycler 480^TM^ platform (Roche Applied Science, Basel, Switzerland) to quantify the mRNA expression levels of target genes. Primers specific for tumor necrosis factor-alpha (TNF-α), interleukin-6 (IL-6), IL-8, IL-16, cyclooxygenase-2 (COX-2), Toll-like receptor 4 (TLR-4), occludin, and claudin were utilized (see Appendix A for detailed sequence information). The expression of each target gene was normalized using housekeeping gene glyceraldehyde 3-phosphate dehydrogenase (GAPDH) (ΔCt = Ct_-Target Gene_ − Ct_-GAPDH_), and relative gene expressions were quantified by the standard 2^−ΔCt^ method.

### 2.10. Statistical Analysis

Data are presented as mean ± standard deviation. Statistical analyses were conducted using the SPSS statistical package (version 18.0; SPSS Inc., Chicago, IL, USA). A one-way analysis of variance (ANOVA) followed by Duncan’s post hoc test was employed to determine significant differences. Statistical significance was set at *p* < 0.05.

## 3. Results

### 3.1. ST Contained a Higher Amount of Total Phenolic Compounds Compared to PS, While PS Exhibited a Higher Concentration of Total Flavonoids

ST demonstrated significantly greater total phenolic compounds, with its value being 1.95 times and 2.08 times higher than those of YT and PS, respectively (*p* < 0.05, Figure 2A). Conversely, PS contained notably higher total flavonoid compounds, surpassing ST and YT by 1.35 times and 2.63 times, respectively (*p* < 0.05, Figure 2B).

### 3.2. ST Demonstrated Superior Antioxidative Effects Compared to YT and PS

ST significantly inhibited DPPH radicals compared to YT (*p* < 0.05, Figure 2C), although its inhibition was only slightly higher than that of PS, with no statistical significance. Additionally, ST significantly reduced ABTS radicals compared to YT and PS (*p* < 0.05, Figure 2D). There was no noticeable difference in ABTS radical scavenging capacity between YT and PS.

### 3.3. ST Inhibited H. pylori Urase More Effectively Compared to YT and PS

At identical concentrations, each herbal formula group exhibited varying levels of ammonia production due to *H. pylori* urase (Figure 3A). For instance, when administered at 25 mg/mL, PS and YT yielded 359 ± 11 and 244 ± 11 ppm of ammonia, respectively. However, ST at the same concentration only resulted in 152 ± 7 ppm of ammonia. Therefore, at a concentration of 25 mg/mL, ST demonstrated a reduction in ammonia production compared to YT and PS, with reductions of 37.7% and 57.7%, respectively.

### 3.4. ST Exerted Higher Anti-H. pylori Activity than YT and PS

The results of the agar diffusion assay indicated that the MIC for each sample was less than 25 mg/mL for ST, less than 50 mg/mL for PS, and approximately 100 mg/mL for YT (Figure 3B). Furthermore, the results of the disk diffusion assay also demonstrated that ST treatment resulted in a larger clear zone compared to YT and PS treatments at identical concentrations (Figure 3C,D). Additionally, ST treatment significantly decreased the ammonia produced by urease from *H. pylori* compared to YT and PS at identical concentrations (*p* < 0.05; Figure 3A).

### 3.5. Three Herbal Formulas Exhibited Potential Anti-Inflammatory Effects in the H. pylori-Infected AGS Cell Model

Treatment with each herbal extract did not decrease AGS cell viability at 24 h up to 100 μg/mL (Figure 4A). However, the combination treatment of herbal extracts with an MOI of 50 of *H. pylori* for 24 h significantly lowered the cell viability compared to treatment with herbal extracts alone (Figure 4A).

After infection with *H. pylori*, the gene expressions of abundant proinflammatory cytokines in AGS cells, including TNF-α, IL-6, COX-2, IL-8, and IL-16, were upregulated significantly (*p* < 0.05, Figure 4C,D). Additionally, the gene expression of TLR-4, a receptor for lipopolysaccharides, was also upregulated by *H. pylori* infection in the AGS cells. After treatment with herbal extracts, all three herbal extracts indicated a significant suppression of the aforementioned proinflammatory cytokine gene expressions (*p* < 0.05, Figure 4C,D). No statistically significant differences in inflammatory cytokine and TLR-4 gene expression were found among the three herbal formulas.

### 3.6. ST and YT Potentially Elevated the Tight Junction Proteins in the H. pylori-Infected AGS Cell Model Rather than PS

*H. pylori-*infected AGS cells exhibited a significant reduction in the gene expression of tight junction proteins such as occludin and claudin compared to non-infected AGS cells (*p* < 0.05; Figure 4B). However, treatment with ST and YT notably elevated the gene expression of these tight junction proteins (*p* < 0.05; Figure 4B), whereas PS treatment did not indicate a significant alteration.

### 3.7. YT Exhibited Greater Morphological Improvement in Vacuolation Compared to ST and PS in the H. pylori-Infected AGS Cell Model

As expected, *H. pylori* infection triggered noticeable morphological changes in AGS cells, including cell rounding, cell membrane ruffling, and vacuolization (Figure 4E). Pre-treatment with the three extracts noticeably protected against *H. pylori*-induced cell damage, as evidenced by a significant reduction in the number of vacuolated cells (*p* < 0.05; Figure 4F). Despite no significant variance in cellular vacuolation among the three formulations, YT treatment indicated a more pronounced decrease in cell vacuolation compared to treatments with ST and PS (Figure 4E).

## 4. Discussion

Oxidative stress arises from an imbalance between the production of free radicals and their scavenging capacity [24]. Many gastrointestinal disorders (GD), such as dyspepsia, gastroesophageal reflux disease (GERD), and inflammatory bowel disease (IBD), are closely linked to chronic inflammation and cellular damage in the gastrointestinal tract due to excessive oxidative stress [25,26]. Recent clinical research has indicated that *H. pylori*, containing the CagA virulence factor, can induce increased levels of reactive oxygen species (ROS) in plasma [27]. Therefore, boosting antioxidative capacity has emerged as a promising therapeutic approach for managing various gastrointestinal diseases [28]. In this study, we initially compared the phenolic and flavonoid contents of the three herbal formulas, as well as their scavenging abilities against free radicals. Formula ST exhibited higher levels of phenolic compounds (>0.8%) compared to YT and PS, while both ST and PS showed higher levels of flavonoid compounds compared to YT. Flavonoids, being a subclass of phenolic compounds due to their similar chemical structures characterized by aromatic rings with hydroxyl groups [29], contribute to the antioxidative capacity. Thus, ST, being rich in phenolic compounds, demonstrated superior free radical scavenging ability. Consequently, we anticipate that ST may have a comparatively greater therapeutic effect on *H. pylori*-related gastritis and dyspepsia compared to YT and PS.

In addition, *H. pylori*, a well-known stomach-residing pathogen, releases various virulence factors, including urease, cytotoxicity-associated immunodominant antigen (CagA), and lipopolysaccharides (LPSs), which induce inflammatory damage in the gastric mucosal layer [30]. Our results from the MIC and disk agar diffusion assays indicated that ST treatment more effectively inhibits *H. pylori* growth compared to YT or PS treatment. The survival ability of *H. pylori* in the acidic stomach environment primarily relies on urease activity [31]. Results from the ammonia assay indirectly suggest that ST treatment leads to a stronger reduction in *H. pylori* urease activity than YT and PS. This finding may be one of the potential mechanisms underlying the anti-*H. pylori* action. However, the high concentrations of herbal formulas observed in vitro, in comparison to AMX, make their direct clinical application unfeasible. Therefore, future studies should include rodent models to ascertain appropriate clinical doses. Furthermore, previous studies have shown that herbs with antioxidative properties can inhibit *H. pylori* growth by scavenging free radicals, which are beneficial for bacterial survival and proliferation [32,33]. Our findings also revealed that superior antioxidative effects were positively correlated with the more potent anti-*H. pylori* activities of the three herbal formulations.

The AGS cell line, derived from human gastric adenocarcinoma, is commonly used in research on gastric cancer and various infections [34]. For instance, AGS cells have been extensively used to establish a simple and efficient in vitro model for assessing interactions between *H. pylori* and the host gastric epithelium [35,36,37]. Therefore, we utilized the *H. pylori*-infected AGS cell model to compare the potential therapeutic effects of the three herbal formulas. As a typical Gram-negative bacterium, *H. pylori* produces LPS, also known as an endotoxin [38]. *H. pylori* endotoxins have been reported to chronically stimulate the production of numerous pro-inflammatory cytokines by gastric epithelial and immune cells through binding to TLR-4 and signal transduction [39]. These persistent inflammatory responses play a crucial role in the pathogenesis of *H. pylori*-associated diseases by contributing to gastric tissue damage [40]. In our study, *H. pylori* infection notably elevated the expression of genes encoding abundant pro-inflammatory cytokines (IL-6, COX-2, IL-8, TNF-α, IL-16) and the LPS receptor (TLR-4) in AGS cells. Conversely, all three herbal formulas significantly reduced these changes, with no apparent differences observed among them. Intriguingly, the effectiveness of these three herbal formulas against *H. pylori* differed from their anti-inflammatory abilities. A probable explanation is that, although herbal formulas possess different antimicrobial abilities, they do not exclusively ameliorate inflammatory damage.

Moreover, Vacuolating Cytotoxin A (VacA) is a protein toxin produced by *H. pylori* which exerts its toxic effects by forming pores in gastric epithelial cells [41]. Previous studies have demonstrated that VacA-induced damage to the gastric epithelium contributes to increased stomach permeability and inflammation [42]. In our study, vacuole formation was more clearly observed in *H. pylori*-infected AGS cells compared to non-infected cells. This disruption contributes to the pathogenesis of *H. pylori*-associated gastric diseases, including gastritis and dyspepsia [43]. Pretreatment with YT resulted in a more pronounced reduction in morphological alterations such as cell membrane ruffling and vacuolization than the other two herbal formulas. The cellular protection effect partially correlated with the increased expression of gap junction protein genes in AGS cells. Our comparative study was limited to in vitro experiments, focusing solely on the interaction between *H. pylori* and the gastric epithelium. In addition, we clarify that the absolute anti-*H. pylori* activity is low compared to antibiotics. The potential gastroprotective efficacy of the herbal formulas might not be primarily due to their direct action against *H. pylori*. Future studies using animal models or humans will elucidate the more intricate interplay between various immune and non-*H. pylori* bacteria, along with their detailed mechanisms.

## 5. Conclusions

Taken together, these three typical Korean herbal formulas collectively demonstrate promising potential in alleviating *H. pylori*-associated gastrointestinal disorders. Proposed mechanisms of action include oxidative stress reduction, the inhibition of *H. pylori* growth and urease activity, and the prevention of damage to gastric epithelial cells (Figure 5). Notably, ST stands out for its effectiveness in reducing oxidative stress, inhibiting *H. pylori* growth, and mitigating inflammation, while YT exhibits superior efficacy in protecting gastric epithelial cells. These preliminary results offer valuable insights for rational clinical applications. Additionally, they lay a solid foundation for future in vivo evaluations aimed at exploring additional mechanisms of action of the three herbal formulas.

## Figures and Tables

**Figure 1 cells-13-00901-f001:**
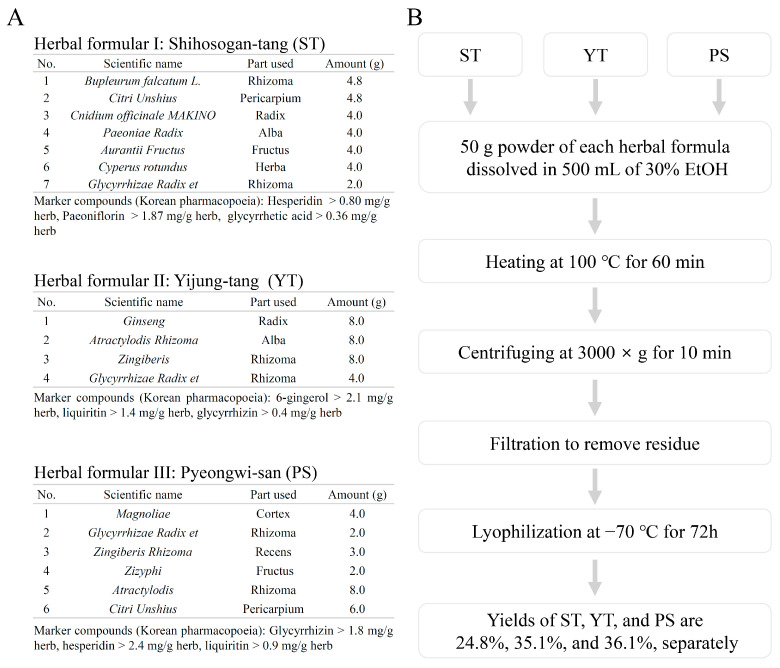
Herbal formula compositions and extraction procedures. (**A**) Detailed components of the three herbal formulas are depicted. (**B**) Extraction processes and yields of the three herbal formulas.

**Figure 2 cells-13-00901-f002:**
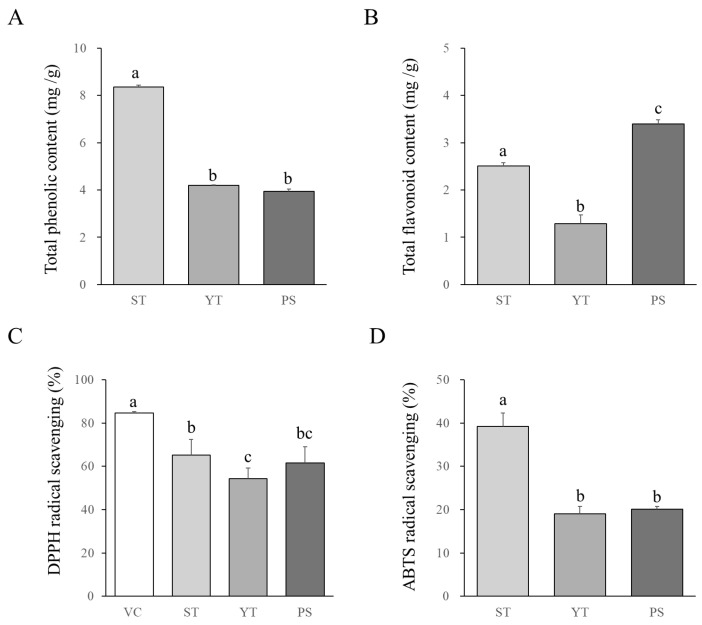
Total phenolic and flavonoid contents and antioxidative activity of the three herbal formulas. Comparison of total phenolic compounds (*n* = 3, **A**) and total flavonoid compounds (*n* = 3, **B**) in the three herbal formulas. The antioxidative capacity of the three herbal formulas was assessed using DPPH (*n* = 3, **C**) and ABTS (*n* = 3, **D**) radical scavenging tests. Data with different letters indicate significant differences (*p* < 0.05) according to our one-way ANOVA.

**Figure 3 cells-13-00901-f003:**
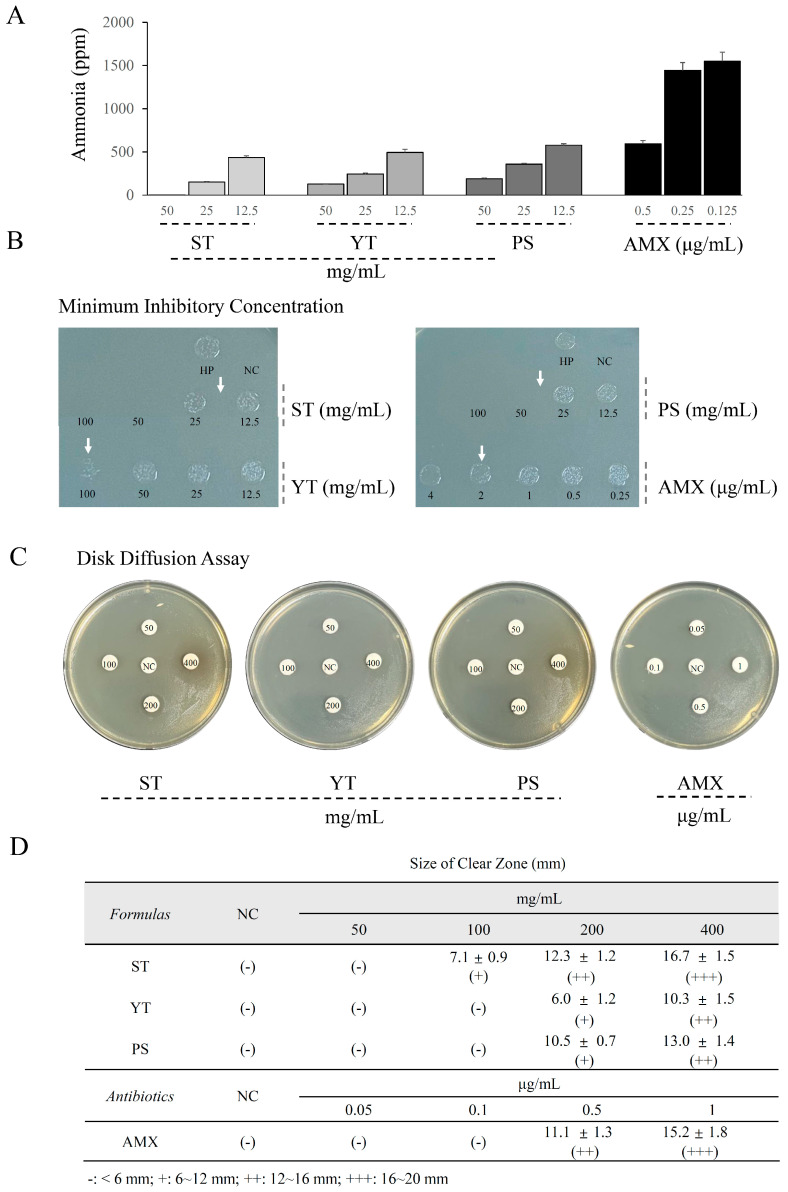
Anti-urease and antimicrobial activities of the herbal formulas against *H. pylori*. (**A**) An ammonia assay was conducted to evaluate *H. pylori* urease activity in vitro (*n* = 3). (**B**) The minimum inhibitory concentration (MIC) for *H. pylori* was determined using the agar dilution method (*n* = 3). The white arrows indicate the range of MIC. (**C**) A disk diffusion assay was applied to evaluate the anti-*H. pylori* capacity (*n* = 3). (**D**) Clear zone sizes were measured using calipers.

**Figure 4 cells-13-00901-f004:**
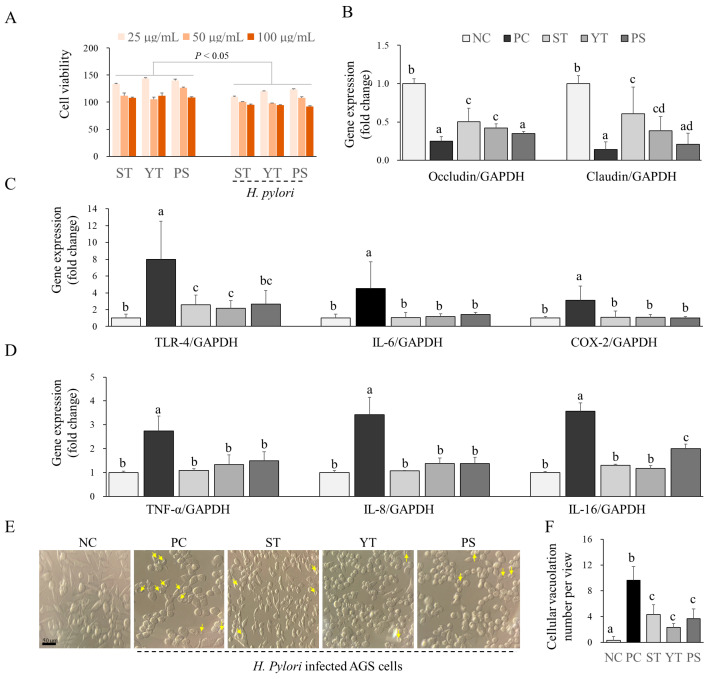
Anti-inflammatory and gastric epithelium-protective effects of the herbal formulas against *H. pylori*. (**A**) AGS cell viability was assessed using the CCK-8 cell proliferation kit (*n* = 3). The relative gene expressions of (**B**) occludin, claudin, (**C**) TLR-4, IL-4, COX-2, and (**D**) TNF-α, IL-8, and IL-16 were determined by real-time PCR (*n* = 3). (**E**) Morphological observations were performed under an optical microscope at 200× magnification (Olympus BX61, Tokyo, Japan). The yellow arrows indicate the cellular vacuolation. (**F**) Cellular vacuolation was counted in five random fields for each sample (*n* = 5). Data with different letters indicate significant differences (*p* < 0.05) according to our one-way ANOVA.

**Figure 5 cells-13-00901-f005:**
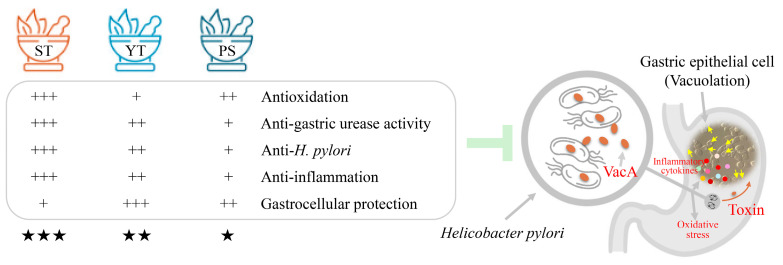
Comparative evaluation of potential therapeutic efficacy of three herbal formulas for *H. pylori*-associated gastrointestinal disorder. Plus signs (+) and stars (★) indicate the relative degree of efficacy.

## Data Availability

All data and related information included in this published paper, including information which could be acquired from the figures, and the data of this study will be made available upon request to the authors without undue reservation.

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
