# Peer review of "Comparative Assessment of the Anti-Helicobacter pylori Activity and Gastroprotective Effects of Three Herbal Formulas for Functional Dyspepsia In Vitro"

_cells, 2024, doi:10.3390/cells13110901_

Round 1

Reviewer 1 Report

Comments and Suggestions for Authors

I read the manuscript of Wang and co-authors with high interest because I truly have found that the paper is interesting, with a good level of novelty and it is particularly attention to the conceptualization and organization.

In detail:

1) I have appreciated the practical aspect of the study of herbal formulation enlisted in official Pharmacopoieas and available in the market (good point!).

2) The introduction combine simplicity, the most important elements and the rationale of the work (very good point!)

3) Methods are in accordance with the most recent literature and controls have been used

4) The discussion is good and adequate

So... why this Reviewer do not accept the paper in the present form?

Because there is a black dot in the paper that makes me doubtful:

The tested concentration are actually 12.5, 25, 50 mg/mL (and higher in some tests)? Considering that the antimicrobial effect of AMX has been confirmed in the range 0.5-2 microg/ml and the correspective human dose is 500 mg twice per day (because of scarce mucosal penetration of all drugs for the stomach), it means that the minimum daily dose of tested samples should be higher than 1 kg!!! In this case the work becomes a nice experimental exercise, but the clinical applications are null.

Also the ascorbic acid (100 mg/mL) served as positive control is used at a senseless concentration.

Did authors miss something and express concentration in a wrong way or this is the fact (please be honest as I am in this note!)

Othe major points 

- In 2.5 please add references or details in order to better understand method and results.

- Figure 3B: AMX concentrations are inverted?

Comments on the Quality of English Language

Good overall quality.

Some typos occur.

Author Response

Revision notes

Reviewer 1

I read the manuscript of Wang and co-authors with high interest because I truly have found that the paper is interesting, with a good level of novelty and it is particularly attention to the conceptualization and organization.

â–¶ I sincerely appreciate the positive and constructive comments. I have carefully addressed each suggestion and made improvements to the manuscript accordingly. Throughout the entire manuscript, all revisions have been highlighted in blue to distinguish them

In detail:

1) I have appreciated the practical aspect of the study of herbal formulation enlisted in official Pharmacopoieas and available in the market (good point!).

2) The introduction combine simplicity, the most important elements and the rationale of the work (very good point!)

3) Methods are in accordance with the most recent literature and controls have been used

4) The discussion is good and adequate

â–¶ Thank you for all the positive comments mentioned above. We are dedicated to simplifying, organizing, and enhancing the readability of the manuscript

So... why this Reviewer do not accept the paper in the present form?

Because there is a black dot in the paper that makes me doubtful:

The tested concentration are actually 12.5, 25, 50 mg/mL (and higher in some tests)? Considering that the antimicrobial effect of AMX has been confirmed in the range 0.5-2 microg/ml and the correspective human dose is 500 mg twice per day (because of scarce mucosal penetration of all drugs for the stomach), it means that the minimum daily dose of tested samples should be higher than 1 kg!!! In this case the work becomes a nice experimental exercise, but the clinical applications are null.

â–¶ Thank you for the essential question. We fully understand the reviewer’s meaning. The test concentration of the formulas was determined based on the results of a cytotoxicity test (data not shown). The maximum concentration that did not exhibit any cytotoxicity in AGS cells was utilized in the present study. Additionally, we believe that in vitro concentrations cannot be directly extrapolated to clinical doses, even when comparing with AMX as a positive control. Furthermore, unlike AMX, herbal formulas contain numerous complex compounds. Hence, for an accurate estimation of the clinical dose, future research should include animal studies. We have addressed this limitation in the "Discussion" section.

As follows: “However, the high concentrations of herbal formulas observed in vitro, in comparison to AMX, make their direct clinical application unfeasible. Therefore, future studies should include rodent models to ascertain appropriate clinical doses.”

Also the ascorbic acid (100 mg/mL) served as positive control is used at a senseless concentration. Did authors miss something and express concentration in a wrong way or this is the fact (please be honest as I am in this note!)

â–¶ Thank you for bringing this to our attention. We sincerely apologize for the mistake regarding the concentration of herbal formulas and vitamin C. Actually, the correct concentration is 100 μg/mL, not 100 mg/mL. We have made the necessary correction in the 'Materials and Methods' section. While the antioxidative effect of herbal formulas may not be stronger compared to ascorbic acid, the main purpose of the present in vitro study was to compare the anti-H. pylori activity and gastroprotective effects among three typical herbal formulas commonly used in clinical practice.

Other major points

- In 2.5 please add references or details in order to better understand method and results.

â–¶ We added the related reference according to the reviewer’s suggestion.

As follows:

  1. Woo, H.J.; Yang, J.Y.; Lee, P.; Kim, J.B.; Kim, S.H. Zerumbone Inhibits Helicobacter pylori Urease Activity. Molecules 2021, 26, doi:10.3390/molecules26092663.

- Figure 3B: AMX concentrations are inverted?

â–¶ We apologize for the error in the concentration sequence. Figure 3B has been corrected.

Reviewer 2 Report

Comments and Suggestions for Authors

The manuscript is interesting, but I think some major revisions are needed.

1) The authors elucidated the compositions of the 3 tested formulations. As regards the extraction technique, they should explain the choice of a decoction. Why this kind of extraction? Is it related to the popular use? Please, discuss.

2) DPPH assay

Usually, more than one concentration is tested, in order to calculate the Ic50 values. Moreover, for plant extracts, lower concentrations are tested (from 1 mg/mL to lower ones).

As the authors wrote: “20 μL of each sample (100 mg/mL) was combined with 80 μL of the DPPH solution in a 96-well plate”, I suppose a concentration equal to 20 mg/mL was tested. It his very high. It is unusual to test such a concentration. I suggest testing lower ones, and  avoiding use this kind of sentences “(from the abstract) with ST excelling in reducing oxidative stress”.

 This is not an excellent antioxidant activity.

3) moreover, in the introduction section, the authors should give more information on the formulations used, their use, their popular use, the biological activity already demonstrated, and the reasons why they decided to test these botanicals.

Author Response

Revision notes

Reviewer 2

The manuscript is interesting, but I think some major revisions are needed.

â–¶ I sincerely appreciate the positive and constructive comments. I have carefully addressed each suggestion, making improvements to the manuscript accordingly. All revisions made throughout the entire manuscript have been highlighted in blue.

1) The authors elucidated the compositions of the 3 tested formulations. As regards the extraction technique, they should explain the choice of a decoction. Why this kind of extraction? Is it related to the popular use? Please, discuss.

â–¶ Thank you for your professional questions. We have briefly discussed the reason for selecting the 30% ethanol extract and have added this information to the “Materials and Methods” section.

As follows: “To ensure standardized comparison and sufficient acquisition of both fat-soluble and water-soluble compounds, we opted the 30% ethanol extract method for the three herbal formulas.”

2) DPPH assay

Usually, more than one concentration is tested, in order to calculate the Ic50 values. Moreover, for plant extracts, lower concentrations are tested (from 1 mg/mL to lower ones).

As the authors wrote: “20 μL of each sample (100 mg/mL) was combined with 80 μL of the DPPH solution in a 96-well plate”, I suppose a concentration equal to 20 mg/mL was tested. It his very high. It is unusual to test such a concentration. I suggest testing lower ones, and  avoiding use this kind of sentences “(from the abstract) with ST excelling in reducing oxidative stress”. This is not an excellent antioxidant activity.

â–¶ Thank you for bringing this to our attention. We sincerely apologize for the mistake regarding the concentration of herbal formulas and vitamin C. Actually, the correct concentration is 100 μg/mL, not 100 mg/mL. We have made the necessary correction in the 'Materials and Methods' section.

While the antioxidative effect of herbal formulas may not be stronger compared to ascorbic acid, the main purpose of the present in vitro study was to compare the anti-H. pylori activity and gastroprotective effects among three typical herbal formulas commonly used in clinical practice. In addition, we deleted “with ST excelling in reducing oxidative stress” in the “Abstract” section according to the reviewer’s suggestion.

3) moreover, in the introduction section, the authors should give more information on the formulations used, their use, their popular use, the biological activity already demonstrated, and the reasons why they decided to test these botanicals.

â–¶ Thank you for the constructive suggestions on improving the quality of the paper. Despite these herbal formulas being commonly used in clinics and covered by Korean National Health Insurance, there have been few studies demonstrating their biological activity thus far. Nonetheless, we have included other related content, including the rationale for drug selection, in the "Introduction" section, following the reviewer's recommendation.

As follows: “In this investigation, we focused on three commonly used oriental herbal formulas covered by the National Health Insurance Service (NHIS, www.nhis.or.kr): Shihosagan-tang (ST, originating from “Jing-Yue-Quan-Shu” in 1624 A.D.), Ijung-tang (YT, originating from “Shang-Han-Lun” in 219 A.D.), and Pyeong-wi-san (PS, originating from Dong-Eui-Bo-Gam in 1613 A.D.). According to traditional medical theory, ST is primarily used for treating stagnation of liver Qi, YT is mainly employed for asthenic symptoms, and PS is largely utilized to address food impaction. Collectively, these formulas are renowned for their clinical efficacy in addressing diverse gastrointestinal ailments. Nevertheless, it remains unclear which formula exhibits a more effective gastroprotective effect and stronger antimicrobial activity against pathogens, such as H. pylori.”

Reviewer 3 Report

Comments and Suggestions for Authors

The supplied manuscript for review details an in vitro study on the anti-Helicobacter pylori activities and gastroprotective effects of three Korean herbal preparations, namely Shihosogan-tang, Yijung-tang, and Pyeongwi-san. The experiments conducted during the study involved several key assessments to evaluate the anti-Helicobacter pylori activity and gastroprotective effects of three Korean herbal formulas including antibacterial testing, antioxidative assessments, cellular experiments, morphological evaluations and gene expression analysis using real-time PCR.

The key findings included:

Shihosogan-tang demonstrated the highest levels of total phenolic compounds, superior antioxidant properties, and the most effective inhibition of H. pylori growth among the three formulas.

Yijung-tang was notably effective in reducing gastric cellular morphological changes, such as vacuolation, compared to Shihosogan-tang and Pyeongwi-san.

All three herbal extracts significantly improved gene expressions of tight junction proteins and reduced pro-inflammatory cytokines in H. pylori-infected AGS cells, indicating potential therapeutic benefits against gastrointestinal disorders associated with H. pylori.

These findings suggest that while each formula has specific strengths, collectively, they offer promising avenues for managing gastrointestinal health impacted by H. pylori .

 The authors should be commended for a well written and presented manuscript which is of a high standard. However, there are a couple of comments/ suggestions that I noted during my review of the manuscript which the authors may wish to address prior to publication.

Introduction – the authors should consider expanding on the history and properties of the three herbal compounds investigated. Suitable background information is referenced in the bibliography but some extra information is warranted including their relevance for inclusion in this study.

Figure 1: Consider revising this to three typed tables (instead of Figure 1 A) and a figure for the process diagram (Figure 1B) or providing a higher resolution format of the images. Text based figures can always lose resolution when exporting

In some instances, H. pylori is not italicised, the authors should review the manuscript and harmonise the formatting throughout the manuscript. See Line 52 and 342

Discussion- There is a difference in the formatting / font size or type in the first paragraph of the discussion Line 276 onwards.

Figure 5:  Consider increasing the contrast of this Figure to make it clearer and use also as graphical abstract as it provides an excellent overview of the conclusions.

Author Response

Revision notes

Reviewer 3

The supplied manuscript for review details an in vitro study on the anti-Helicobacter pylori activities and gastroprotective effects of three Korean herbal preparations, namely Shihosogan-tang, Yijung-tang, and Pyeongwi-san. The experiments conducted during the study involved several key assessments to evaluate the anti-Helicobacter pylori activity and gastroprotective effects of three Korean herbal formulas including antibacterial testing, antioxidative assessments, cellular experiments, morphological evaluations and gene expression analysis using real-time PCR.

The key findings included:

Shihosogan-tang demonstrated the highest levels of total phenolic compounds, superior antioxidant properties, and the most effective inhibition of H. pylori growth among the three formulas.

Yijung-tang was notably effective in reducing gastric cellular morphological changes, such as vacuolation, compared to Shihosogan-tang and Pyeongwi-san.

All three herbal extracts significantly improved gene expressions of tight junction proteins and reduced pro-inflammatory cytokines in H. pylori-infected AGS cells, indicating potential therapeutic benefits against gastrointestinal disorders associated with H. pylori.

These findings suggest that while each formula has specific strengths, collectively, they offer promising avenues for managing gastrointestinal health impacted by H. pylori .

The authors should be commended for a well written and presented manuscript which is of a high standard. However, there are a couple of comments/ suggestions that I noted during my review of the manuscript which the authors may wish to address prior to publication.

â–¶ I sincerely appreciate the positive and professional comments. I have carefully addressed each suggestion, making improvements to the manuscript accordingly. All revisions made throughout the entire manuscript have been highlighted in blue.

Introduction – the authors should consider expanding on the history and properties of the three herbal compounds investigated. Suitable background information is referenced in the bibliography but some extra information is warranted including their relevance for inclusion in this study.

â–¶ Thank you for your comments. I have included additional information regarding the difference among the three herbal formulas in the "Introduction" section.

As follows: "In this investigation, we focused on three commonly used oriental herbal formulas cov-ered by the National Health Insurance Service (NHIS, www.nhis.or.kr): Shihosagan-tang (ST, originating from “Jing-Yue-Quan-Shu” in 1624 A.D.), Ijung-tang (YT, originating from “Shang-Han-Lun” in 219 A.D.), and Pyeong-wi-san (PS, originating from Dong-Eui-Bo-Gam in 1613 A.D.). According to traditional medical theory, ST is primarily used for treating stagnation of liver Qi, YT is mainly employed for asthenic symptoms, and PS is largely utilized to address food impaction. Collectively, these formulas are re-nowned for their clinical efficacy in addressing diverse gastrointestinal ailments. Never-theless, it remains unclear which formula exhibits a more effective gastroprotective effect and stronger antimicrobial activity against pathogens, such as H. pylori.”

Figure 1: Consider revising this to three typed tables (instead of Figure 1 A) and a figure for the process diagram (Figure 1B) or providing a higher resolution format of the images. Text based figures can always lose resolution when exporting

â–¶ Thank you for your helpful comment. I have increased the resolution of Figure 1.

In some instances, H. pylori is not italicised, the authors should review the manuscript and harmonise the formatting throughout the manuscript. See Line 52 and 342

â–¶ Thank you for pointing it out. I corrected it carefully.

Discussion- There is a difference in the formatting / font size or type in the first paragraph of the discussion Line 276 onwards.

â–¶ I corrected it meticulously.

Figure 5:  Consider increasing the contrast of this Figure to make it clearer and use also as graphical abstract as it provides an excellent overview of the conclusions.

â–¶ Thank you for your suggestion. I have enhanced the contrast of the Figure 5 and have chosen to utilize it as our graphical abstract.

Round 2

Reviewer 1 Report

Comments and Suggestions for Authors

Dear authors,

thank you for your replies. Please excuse me, but I can't still understand the used concentrations because in figure 4 you reported the concentrations as microg/ml, not mg/ml but you replied that in antibacterial essays you used 100 mg/ml; I can assure you that 100 mg/ml of any herbal extract is toxic in cells (it is a 10% in water!!!).

Not only... I have this big doubt because 100 mg/ml is the concentration used to have the extraction (50 g of drugs in 500 ml), a very very high concentration of the crude extract, cited to frame the situation. But now, following your text... you tell me that you diluted the extract, dried and resuspended, in medium in multiwells for anti-HP test or cell viability (and normally the dilution is 1:10 or 1:100 to not alter medium composition): is it correct? It means that your new extract concentration was 1 g/ml or 10 g/ml and I struggle to understand how you have obtained the solubility of these preparations!!!!!!  

Is all this correct or I or you miss a passage?

Thank you colleagues for your reply.

Comments on the Quality of English Language

Minor revisions required

Author Response

Revision Note

Dear authors,

Thank you for your replies. Please excuse me, but I can't still understand the used concentrations because in figure 4 you reported the concentrations as microg/ml, not mg/ml but you replied that in antibacterial essays you used 100 mg/ml; I can assure you that 100 mg/ml of any herbal extract is toxic in cells (it is a 10% in water!!!).

Not only... I have this big doubt because 100 mg/ml is the concentration used to have the extraction (50 g of drugs in 500 ml), a very very high concentration of the crude extract, cited to frame the situation. But now, following your text... you tell me that you diluted the extract, dried and resuspended, in medium in multiwells for anti-HP test or cell viability (and normally the dilution is 1:10 or 1:100 to not alter medium composition): is it correct? It means that your new extract concentration was 1 g/ml or 10 g/ml and I struggle to understand how you have obtained the solubility of these preparations!!!!!! 

Is all this correct or I or you miss a passage?

Thank you colleagues for your reply.

â–¶ We are very impressed by the reviewer's professionalism and grateful for catching the crucial point that we overlooked. We sincerely apologize for the unit errors scattered throughout the manuscript, making it hard for the reviewer to understand.

After reviewing the experimental records, we found that the concentration of the herbal formula extract used in AGS cell experiments is μg/mL, NOT mg/mL. The concentrations of herbal formulas (100 μg/mL) in co-culture cell model refer to the final concentration, NOT the treatment concentration. We have revised the unit and clarified it in the “Materials and Methods” section (line 96-97).

Regarding the antibacterial assays, we indeed used mg/mL, not μg/mL. Of course, we agree that the antibacterial effect of the three herbal formulas is quite low as compared to antibiotics. However, in the present study, we only focused on comparing among the commonly clinical used three herbal formulas. We additionally added the corresponding limitation in the “Discussion” section in order to avoid misunderstanding. As follows: “In addition, we clarify that the absolute anti-H. pylori activity is low compared to antibiotics. The potential gastroprotective efficacy of the herbal formulars might be not primarily due to their direct action against H. pylori.” (line 347-349)

Once again, we honestly appreciate your patience and help in improving quality of the manuscript.

Reviewer 2 Report

Comments and Suggestions for Authors

Dear Editors, the authors addressed my suggestions and comments. I suggest accepting the revised version of the paper. 

Best regards

Author Response

Thank you for your positive feedback.

Round 3

Reviewer 1 Report

Comments and Suggestions for Authors

Dear authors, thank you for your helpful feedback and for having fixed all major concerns.

Please read carefully the final draft, some typos still remain.

Best wishes.